# Effects of Ankle Position While Performing One- and Two-Leg Floor Bridging Exercises on Core and Lower Extremity Muscle Recruitment

**DOI:** 10.3390/bioengineering11040356

**Published:** 2024-04-05

**Authors:** Rafael F. Escamilla, Irwin S. Thompson, Joe Carinci, Daniel MacLean, Lisa MacLean, Arnel L. Aguinaldo

**Affiliations:** 1Department of Physical Therapy, California State University, Sacramento, CA 95819, USA; irwin.thompson@csus.edu; 2UC Davis Sports Medicine & Orthopedics, Sacramento, CA 95817, USA; jmcarinci@ucdavis.edu; 3MAC Performance Physical Therapy, Sacramento, CA 95827, USA; dan@macperformancept.com (D.M.);; 4Department of Kinesiology, Point Loma Nazarene University, San Diego, CA 92106, USA; arnelaguinaldo@pointloma.edu

**Keywords:** EMG, lower back pain, lumbar spine, bridge

## Abstract

Given there are no known studies which have examined multiple lower extremity muscles between different ankle positions during bridging activities, the objective was to assess how employing two different ankle positions (PF versus DF) while performing five common bridging exercises (three bipedal and two unipedal) used in rehabilitation and athletic performance affect core and select lower extremity muscle EMG recruitment. Twenty healthy subjects performed a 5 s isometric hold during five two- and one-leg bridge exercises: (1) on right leg with left knee to chest (1LB-LFlex); (2) on right leg with left knee extended (1LB-LExt); (3) standard two-leg bridge (2LB); (4) two-leg bridge with resistance band around knees (2LB-ABD); and (5) two-leg bridge with ball between knees (2LB-ADD). Surface electromyographic (EMG) data were collected using a Noraxon Telemyo Direct Transmission System from fourteen muscles: (1) three superficial quadriceps (VM, VL, and RF); (2) three hip abductors (TFL, GMED, and GMAX); (3) medial hamstrings (ST) and lateral hamstrings (BF); (4) hip adductors (ADD); (5) erector spinae (ES); (6) latissimus dorsi (LATS); (7) upper rectus abdominis (RA); and (8) external oblique (EO) and internal oblique (IO). EMG data were normalized by maximum voluntary isometric contractions (MVICs). A paired *t*-test (*p* < 0.01) was used to assess differences in normalized mean EMG activities between DF and PF for each exercise. EMG activities were significantly greater in DF than PF for the (a) VM, VL, and RF during 1LB-LFlex; (b) ADD during 1LB-LFlex, 1LB-LExt; (c) EO during 1LB-LFlex; and (d) IO during 1LB-LFex. In contrast, EMG activities were significantly greater in PF than DF for ST and BF during all five bridge exercises. Bridging with PF (feet flat) was most effective in recruiting the hamstrings, while bridging with DF (feet up) was most effective in recruiting the quadriceps, hip adductors, and internal and external obliques.

## 1. Introduction

Lumbopelvic core training remains a key element in rehabilitation and performance exercise protocols. The lumbopelvic core is a complex of deep and superficial lumbopelvic hip muscles. Deep core muscles include transversus abdominis, internal obliques, transversospinalis, quadratus lumborum, and psoas major and minor. Superficial core muscles include rectus abdominis, external oblique, erector spinae, latissimus dorsi, gluteus maximus and medius, hamstrings, hip adductors, and rectus femoris [1,2]. Optimally synchronized core muscle recruitment is important in many functional and athletic activities to promote proximal stability, which facilitates distal mobility. Specifically, smaller, deeper core muscles and larger, superficial core muscles must contract in sequence with appropriate timing and tension [2].

Weakness in core musculature has been associated with a variety of pathologies. Notably, gluteal weakness and discoordination is a risk factor for acetabular labral and anterior cruciate ligament (ACL) tears [3], gluteal tendinopathy [4], hip osteoarthritis (OA) [5], patellofemoral pain syndrome (PFPS) [6], iliotibial (IT) band syndrome [7], and ankle instability [8]. Additionally, lumbopelvic hip disorders are linked to poor neuromuscular control of the core musculature [9].

Floor bridging exercises have been identified for their utility in targeting the erector spinae and abdominal obliques [10], the gluteus maximus and medius [11], the quadriceps, hamstrings, and hip adductors [12]. Bridging exercises also enhance contractions from the lumbar multifidi, erector spinae, external obliques, and rectus abdominis, in hierarchical order [13]. Strengthening the lumbopelvic hip complex has been shown to decrease lower extremity injury risk and enhance performance [3,14], as well as reduce the risk of lumbar spine injuries by enhancing spinal stability [1]. Consequently, rehabilitation professionals are regularly tasked with prescribing strengthening exercises designed to obviate potential disorders or redress existing ones. Comparisons of exercise type have revealed that bipedal bridging elicits roughly equal ratios of activation between the internal and external obliques, whereas unipedal bridging favorably recruits the ipsilateral internal oblique as compared to the ipsilateral external oblique [10]. Unipedal bridging has also been shown to recruit greater activation of the ipsilateral gluteus medius [11].

The standard supine two-leg (bipedal) and one-leg (unipedal) bridge exercises are commonly utilized in the rehabilitation of patients with lower extremity and lumbar pathology [15], particularly as they exist in a sequential middle ground between non-weight-bearing and weight-bearing exercises, and are also commonly utilized by athletes in sport for the development of core musculature and sport performance enhancement [16,17]. Bipedal and unipedal bridging are also commonly performed with exercise variations, including resisted isometric hip abduction and adduction (bipedal), and contralateral lower extremity flexion or extension (unipedal) [16,17]. Moreover, it is commonly believed that performing these bridging exercises with the feet up and only the heels on the ground (ankle dorsiflexion, DF) recruits the gluteus maximus to a greater extent compared to bridging with the feet flat on the ground (ankle plantarflexion, PF), and that bridging with PF recruits the hamstrings to a greater extent compared to bridging with DF [16,17]. However, these beliefs have never been scientifically validated. The only known study that examined lower extremity muscle activity while bridging with different foot positions was performed by Yoo [18], who examined hamstring and gluteus maximus activity between traditional two-leg bridging with the feet flat versus two-leg bridging with the heels off the ground. Yoo [18] reported that hamstring activity was significantly less and gluteus maximus activity was significantly greater when bridging while raising the heels off the ground compared to bridging with the feet flat on the ground. However, there are no known studies that have examined hamstring and gluteus maximus activity, as well as additional lower extremity muscles, while bridging on the heels with the toes up (DF ankle position) instead of on the toes with the heels up. Therefore, the purpose of this study was to assess how employing two different ankle positions (PF versus DF) while performing five common bridging exercises (three bipedal and two unipedal) [16,17] used in rehabilitation and athletic performance affect core and select lower extremity muscle EMG recruitment. It was hypothesized that gluteus maximus, quadriceps, and abdominal/oblique muscle EMG activities would be greater when performing bridging exercises with DF compared to PF, while hamstring muscle EMG activities would be greater when performing bridging exercises with PF compared to DF.

## 2. Methods

### 2.1. Participants

Twenty healthy, young volunteers served as subjects: ten males and ten females. Mean (SD) age, mass, and height were 24.4 (1.5) years, 59.7(7.5) kg, and 164.0 (6.7) cm, respectively, for females, and 24.9 (1.6) years, 78.5 (7.9) kg, and 176.3 (4.7) cm, respectively, for males. To optimize the quality of the electromyographic (EMG) signal collected, this study was limited to a convenience sample of 20 healthy, young subjects (10 male and 10 female) who had normal or below normal body fat for their age group, in accordance with standards set by the American College of Sports Medicine [19]. Baseline skinfold calipers (Model 68900, Country Technology, Inc., Gays Mill, WI, USA) and appropriate regression equations were used to assess percent body fat. Mean (SD) percent body fat was 18.2 (2.4)% for females and 11.7 (3.1)% for males. All subjects provided written informed consent in accordance with the Institutional Review Board at California State University, Sacramento. Individuals were excluded from the study if they had any musculoskeletal pathologies, as assessed by a licensed physical therapist, that prevented them from being able to perform all exercises pain-free, through their full range of motion, and with proper form and technique. All subjects were also excluded from the study if they did not have at least 3 years’ experience in performing bipedal and unipedal floor bridging exercises, and were excluded if their body fat was above normal, as described previously.

### 2.2. Exercise Descriptions

The starting position for the three 2-leg floor bridge exercises was the supine hook-lying position with the hips flexed approximately 50°, the knees flexed approximately 100°, and the separation between both feet and between both knees was hip width distance, allowing both lower extremities to stay parallel with each other throughout the floor bridge exercises. Both arms were positioned next to the body with the palms down. The subject pushed through the feet and hands, lifting the buttocks upwards until the hips were in a neutral position with 0° hip flexion, with the knees, hips, and shoulders approximately in a straight line. This ending position for the three 2-leg floor bridge exercises was performed with the two ankle positions (Figure 1, Figure 2 and Figure 3): (a) PL—both feet remained flat on the floor, and (b) DF—both feet were lifted up with maximum ankle DF, where only the heels remained on the ground. The difference between the three 2-leg floor bridge exercises was that one exercise was performed with no external resistance (2LB, Figure 1a,b), one exercise was performed with an 8-inch Theraband^®^ resistance band (Theraband, Akron, OH, USA—bands of different resistance were available) around the distal thighs with a perceived exertion of effort from the band of “somewhat hard”, which was 13–14 on a 6–20 rating of perceived exertion scale (2LB-Abd, Figure 2a,b), and one exercise was performed with a 21.6 cm diameter 2-ply rubber ball (Model #SP85R, Tachikara USA Inc., Sparks, NV, USA) inflated to 1.5 PSI and placed between the knees and squeezed until a perceived exertion of effort of “somewhat hard” was achieved (2LB-Add, Figure 3a,b). Each subject held these ending positions for the three 2-leg bridging exercises for 5 s while EMG data were collected.

The starting and ending positions for the two 1-leg floor bridge exercises were initially the same as the 2-leg floor bridge exercises, but once the hips were straight with 0° hip flexion, with the knees, hips, and shoulders approximately in a straight, the following changes occurred for the remaining three 1-leg bridge exercises. (1) The left hip and knee flexed and the left knee was pulled toward the chest just hard enough to keep in place a tennis ball-sized ball positioned on the lower left ribs. The right foot either stayed flat on the floor (PF) or was maximally lifted up (DF) with right heel only on the ground (1LB-LFlex, Figure 4a,b). (2) the left knee fully extended and the right foot either stayed flat on the floor (PF) or was maximally lifted up (DF) with the right heel only on the floor (1Lb-LExt, Figure 5a,b).

### 2.3. Procedures

During a pre-test session that took place 1 week prior to the testing session, each subject practiced all exercises as previously described and determined what Theraband^®^ resistance band was appropriate. During the pre-test session, each subject received instructions from a physical therapist, who explained and demonstrated proper execution of each exercise. During the pre-test, each participant’s body fat was also assessed as previously described.

Each subject arrived at the Biomechanics Laboratory for testing and changed into appropriate workout attire for testing. Blue Sensor (Ambu Inc., Linthicum, MD, USA) disposable surface electrodes (type M-00-S) were used to collect EMG data. These oval-shaped electrodes (22 mm wide and 30 mm long) were placed in a bipolar configuration along the longitudinal axis of each muscle, with a center-to-center distance of approximately 3 cm between electrodes. Prior to applying the electrodes, the skin was prepared by shaving, abrading, and cleaning with isopropyl alcohol wipes to reduce skin impedance. Electrode pairs were then placed on the subject’s right side (arbitrarily chosen) for the following superficial muscles in accordance with procedures previously described [20,21,22,23]: (a) RA = upper rectus abdominis; (b) EO = external oblique; (c) IO = internal oblique; (d) LATS = latissimus dorsi; (e) ES = lumbar paraspinals (erector spinae); (f) RF = rectus femoris; (g) VL = vastus lateralis; (h) VM = vastus medialis; (i) TFL = tensor fascia latae; (j) ADD = hip adductors (primarily adductor longus); (k) GMED = gluteus medius; (l) GMAX = gluteus maximus; (m) ST = medial hamstrings (semimembranosus and semitendinosus, but primarily semitendinosus); and (n) BF = lateral hamstrings (biceps femoris). A ground (reference) electrode was positioned over the skin of the right acromion process. Electrode cables were connected to the electrodes and taped to skin appropriately to minimize pull on the electrodes and movement of the cables.

Electrodes were only positioned over muscles on one side of the body because symmetry was assumed for the 2-leg bridge exercises, with muscle activity on the right side reflective of muscle activity on the left side. Left- and right-side EMG symmetry has been demonstrated in core muscles during bipedal (2-leg) supine and prone position exercises similar to the bipedal bridge [24]. However, because muscle symmetry cannot be assumed for left and right sides of the body for 1-leg bridge exercises, two of the 1-leg bridge exercises were performed on opposite legs and were mirror reflections of each other, and 1LB-LExt with the right foot on the ground and the left leg extended above the ground (Figure 5a,b). Consequently, EMG activities on the right side of the body during 1LB-RExt should be representative and similar to EMG activities on the left side of the body during 1LB-LExt. In effect, this would be similar to having electrodes on both sides of the body during 1LB-LExt.

Once electrodes were positioned, the subject warmed up and practiced the exercises as needed, and then data collection commenced. EMG data were sampled at 1000 Hz using a Noraxon Telemyo Direct Transmission System (Noraxon USA, Inc., Scottsdale, AZ, USA). The EMG amplifier bandwidth frequency was 10–500 Hz with an input impedance of 20,000 kΩ, and the common-mode rejection ratio was 130 dB.

As previously described [20,21,25], EMG data from each muscle tested were first collected during two 5 s maximum voluntary isometric contractions (MVICs) to normalize the EMG data collected during the exercises. Each subject was given verbal encouragement for each MVIC to help ensure a maximum effort throughout the 5 s duration, and the subject was asked after each MVIC if they felt it was a maximum effort. If not, the MVIC was repeated. Approximately 1 min rest was given between each MVIC, and approximately 2 min rest was given between each exercise trial. Subsequent to the MVICs, EMG data were collected during a 5 s isometric contraction during the end positions shown for each exercise in Figure 1, Figure 2, Figure 3, Figure 4 and Figure 5. All MVICs and exercises were first randomized, and then to counterbalance the repeated measures design and minimize the risk of an order effect, half of the subjects performed all MVICs and exercises in the randomized order and the other half of the subjects performed all MVICs and exercises in the reverse order.

### 2.4. Data Processing

Raw EMG signals were processed using Noraxon Myoresearch (Noraxon USA, Inc., Scottsdale, AZ, USA) and were full-wave rectified, smoothed with a 10 ms moving average window and linearly enveloped, and then averaged over the entire 5 s duration of each exercise performed. For each repetition, the EMG data were normalized for each muscle and expressed as a percentage of a subject’s highest corresponding MVIC trial, which was determined by calculating throughout the 5 s MVIC the highest average EMG signal over a 1 s time interval. Normalized EMG data were then used in statistical analyses.

### 2.5. Data Analysis

A paired *t*-test was employed to assess differences in normalized mean EMG muscle activities between DF and PF for each exercise. The level of significance used was *p* < 0.01.

## 3. Results

Significant differences (*p* < 0.01) were observed in normalized mean EMG activity between ankle plantar flexion (PF) and ankle dorsiflexion (DF) among the five exercises (Table 1). EMG activity was significantly greater in DF than PF for the (a) quadriceps musculature (VM, VL, and RF) during both one-leg bridge exercises (1LB-LFlex and 1LB-LExt); (b) hip adductors (ADD) during both one-leg bridge exercises; (c) external obliques (EO) during 1LB-LFlex; and (d) internal obliques during 1LB-LFex. In contrast, EMG activity was significantly greater in PF than DF for the hamstrings (both ST and BF) during all three two-leg bridge exercises and both one-leg bridge exercises.

## 4. Discussion

The current study investigated the effects of two different ankle positions (DF and PF) on core (lumbopelvic hip complex) and select lower extremity musculature during common variations of bipedal and unipedal bridging exercises. As hypothesized, greater hamstring muscle recruitment occurred with unipedal and bipedal bridging with PF compared to unipedal and bipedal bridging with DF. However, our hypothesis that gluteus maximus, quadriceps, and abdominal musculature would be significantly greater with DF compared to PF was only partially substantiated. Most notable was that the gluteus maximus, contrary to common beliefs [16,17], did not exhibit significantly greater muscle activity with DF compared to PF. Instead, hip adductor EMG was significantly greater with DF than PF for both one-leg bridging exercises and for one out of three of the two-leg bridge exercises, and quadriceps (VM, VL, and RF) EMGs were significantly greater with DF than PF but only for the two one-leg bridging exercises. Lastly, as hypothesized, abdominal EMGs were significantly greater with DF than PF but only for the IO and EO and not for RA, and only for select exercises.

Notably absent in the literature is the effect of ankle position on muscle activation during bridging, and the current study is the only known study that examined the effects of ankle DF and PF during floor bridging on core and select lower extremity EMG activity. The only other study that examined lower extremity muscle activity during the bridge using different ankle positions was conducted by Yoo [18], who reported that hamstring activity was significantly less when bridging while raising the heels off the ground (26.9 ± 5.2%) compared to bridging with the feel flat on the ground (31.3 ± 6.9%), while gluteus maximus activity was significantly greater when bridging while raising the heels off the ground (25.6 ± 7.2%) compared to bridging with the feel flat on the ground (20.3 ± 5.6%). While these results cannot be compared to the results of the current study because ankle positions were partially different (heels up instead of toes up), the greater hamstring activity than gluteus maximus activity while bridging with the feet flat is similar between studies, and the EMG magnitudes are also similar between studies. Several authors have explored the role that ankle position plays on lower extremity and trunk muscle peak torque generation in other types of exercise. For instance, isokinetic knee flexion and extension peak torque has been shown to increase with active ankle dorsiflexion (DF) [26,27]. Additionally, DF produced greater transversus abdominis thickness, as recorded by ultrasound, during the abdominal draw-in maneuver [28]. Moreover, it was found by Chen et al. [29,30] that all ankle positions, apart from neutral, contributed to greater isometric pelvic floor muscle contractions. Notably, and in contrast to other studies, the authors reported that ankle plantar flexion (PF) was shown to elicit a greater response in associated muscles.

There were several muscles that were not affected by varying ankle position. Performing two-leg and one-leg floor bridging exercises with either DF or PF produced similar muscle EMG patterns for the gluteus medius, gluteus maximus, tensor fascia latae, lumbar paraspinals (erector spinae), latissimus dorsi, and upper rectus abdominis. Therefore, when the goal is to target and strengthen these muscles (largely hip abductors, trunk flexors and extensors, and hip extensors), it appears that floor bridging with either DF or PF could be equally effective. However, when the goal is to target and strengthen the quadriceps, hip adductors, and trunk rotators, performing the floor bridge with DF may be more beneficial than performing the floor bridge with PF. In contrast, when the goal is to target and strengthen the hamstrings (knee flexors and hip extensors), performing the floor bridge with PF may be more beneficial than performing the floor bridge with DF.

The prevailing theory supporting muscle activation changes with altered ankle positions is based on the proprioceptive neuromuscular facilitation principle of irradiation—or excitation overflow—which contributes to induced temporal and spatial summation of contractions of the proximal and distal adjacent muscles [28]. This phenomenon is well-established and has been reported in the historical literature [31].

Gontijo et al. [32] examined proximal-to-distal irradiation by comparing DF and PF strength when coupled with resisted trunk flexion and extension and some of their results were similar to the current study. Greater DF strength coincided with active trunk flexion, and PF strength was enhanced during active trunk extension. The belief is that irradiated muscle recruitment resembles primitive movement patterns associated with ambulation. Specifically, during the swing phase of gait, knee extension and hip flexion are associated with DF, whereas hip extension and knee flexion are more closely aligned with PF. The results presented in Table 1 underscore this relationship, with quadricep activity greater in DF than PF, and hamstring activity greater in PF than DF. Therefore, clinicians may find utility in incorporating these ankle variations when administering bridging exercises to patients who require quadricep or hamstring strengthening, as well as strengthening of other musculature.

## 5. Limitations

Potential limitations of this study include a relatively small sample size (n = 20), a relatively homogeneous sample of young, healthy adults, and the potential for cross-talk from neighboring muscles from surface EMG. However, careful electrode placement that adhered to established protocols was employed to mitigate such outcomes. The electrode placements utilized have been shown to minimize EMG cross-talk from other muscles [23]. This is especially true for the internal oblique, which was the only muscle tested that was not superficial. Because the internal oblique is deep to the external oblique, it is susceptible to considerable EMG cross-talk from the external oblique. However, it has been demonstrated that the internal oblique is only covered by the aponeurosis of the external oblique, and not the external oblique muscle, within the triangle confined by the inguinal ligament, lateral border of the rectus sheath, and a line connecting the ASISs [23]. Therefore, surface electrodes are appropriate to use for the internal oblique when electrode placement is within this area, especially when clinical questions are being discussed and if a small percentage of EMG cross-talk is acceptable. In fact, when performing exercises similar to those in the current study, mean internal and external oblique EMG data from surface electrodes (similarly located as in the current study) were only approximately 10% different compared to mean internal and external oblique EMG data from intramuscular electrodes [22]. McGill et al. [22] have concluded that appropriately placed surface electrodes accurately reflect (within 10%) the muscle activity within the internal or external oblique muscles.

## 6. Conclusions

This study has highlighted differences in lumbopelvic hip complex muscle recruitment when performing bipedal and unipedal bridging variations with ankle dorsiflexion and plantar flexion. When the goal is to maximize hamstring muscle recruitment, plantar flexion is more effective than dorsiflexion. Conversely, when the goal is to maximize quadriceps, hip adductor, and abdominal oblique muscle recruitment, dorsiflexion is more effective than plantar flexion. These findings can guide the clinician in prescribing exercises for enhancing core stability and coordination, and also help athletes in training the core and lower extremity musculature.

## Figures and Tables

**Figure 1 bioengineering-11-00356-f001:**
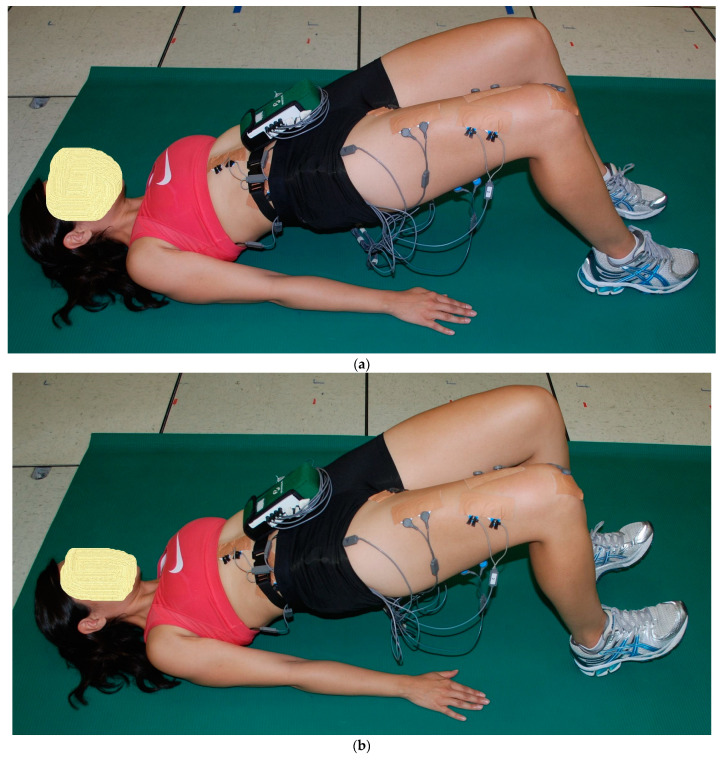
(**a**) (top) and (**b**) (bottom). Two-leg bridge (2LB) with (**a**) both feet flat (ankle plantar flexion, PF) and (**b**) both feet up (ankle dorsiflexion, DF).

**Figure 2 bioengineering-11-00356-f002:**
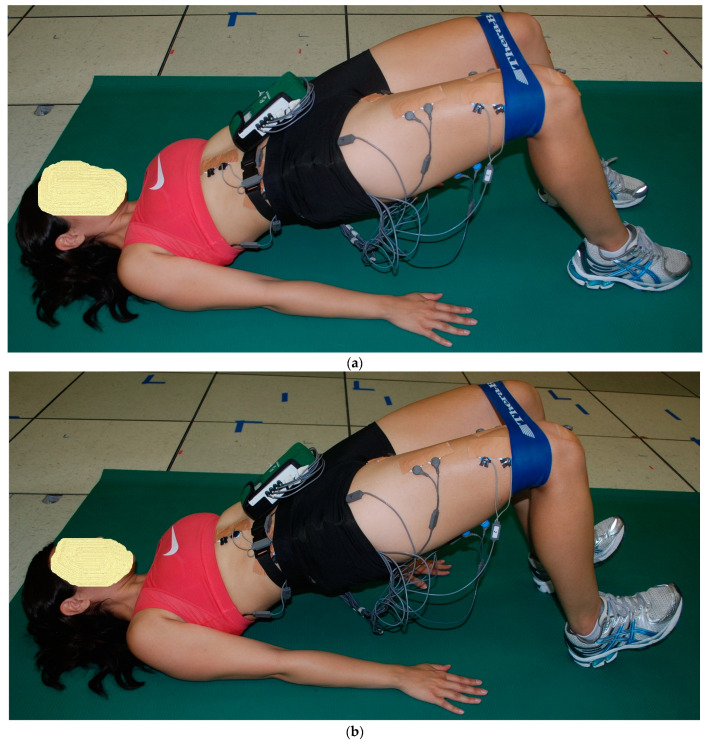
(**a**) (top) and (**b**) (bottom). Two-leg bridge with resistance band around knees (2LB-ABD) with (**a**) both feet flat (ankle plantar flexion, PF) and (**b**) both feet up (ankle dorsiflexion, DF).

**Figure 3 bioengineering-11-00356-f003:**
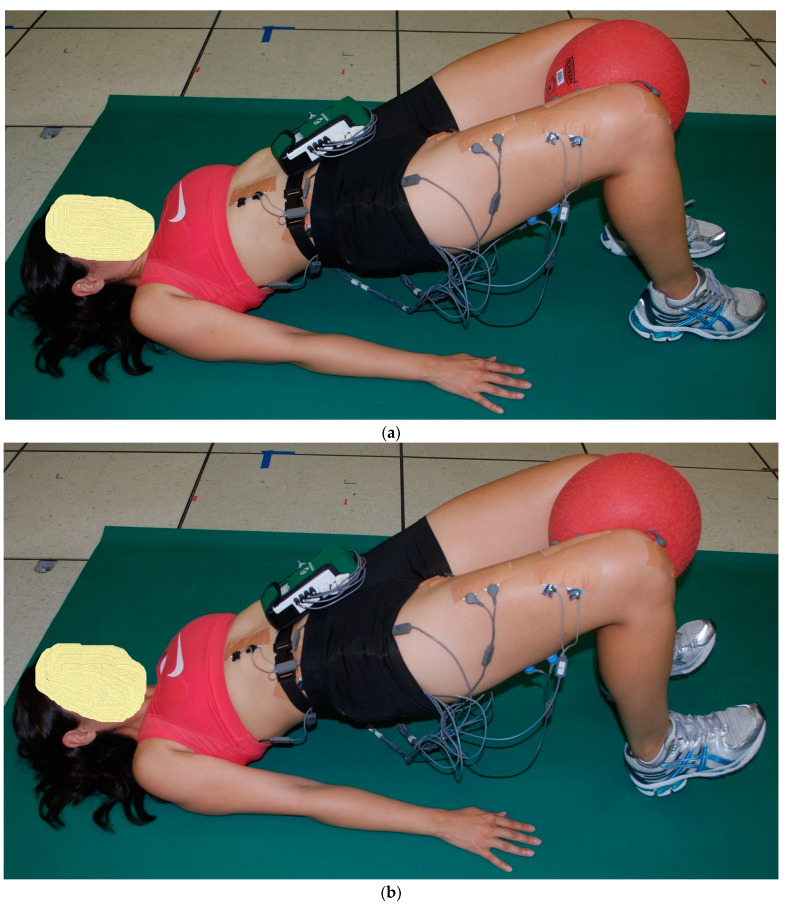
(**a**) (top) and (**b**) (bottom). Two-leg bridge with ball between knees (2LB-ADD) with (**a**) both feet flat (ankle plantar flexion, PF) and (**b**) both feet up (ankle dorsiflexion, DF).

**Figure 4 bioengineering-11-00356-f004:**
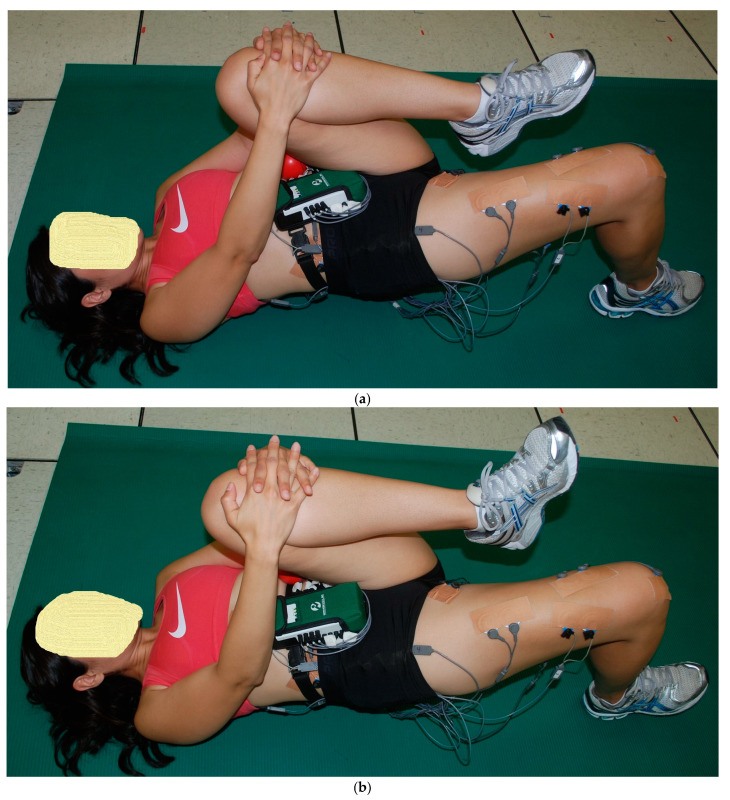
(**a**) (top) and (**b**) (bottom). One-leg bridge with left knee flexed to chest (1Lb-LFlex) with (**a**) right foot flat (right ankle plantar flexion, PF) and (**b**) right foot up (right ankle dorsiflexion, DF).

**Figure 5 bioengineering-11-00356-f005:**
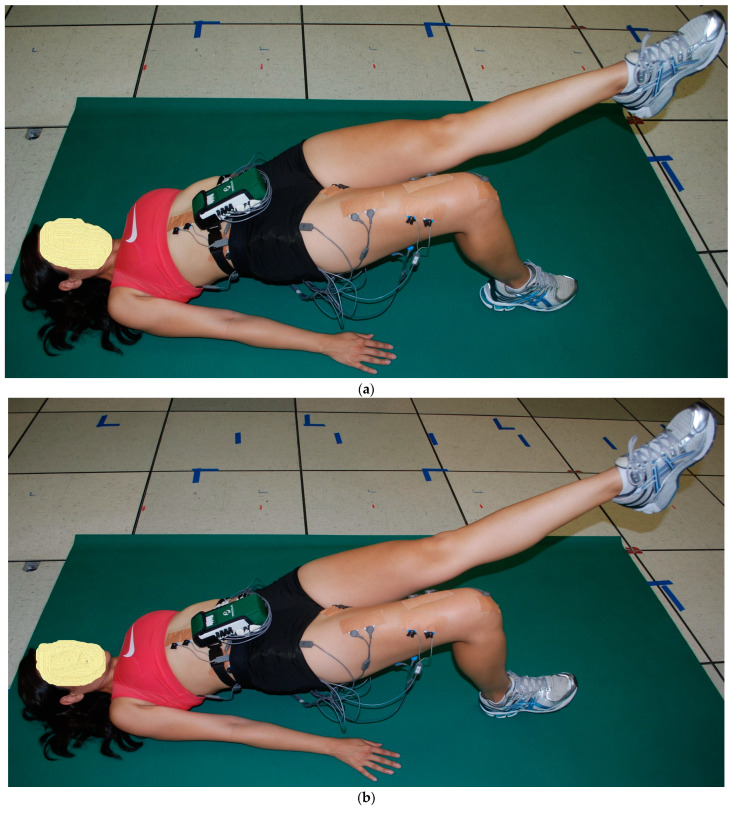
(**a**) (top) and (**b**) (bottom). One-leg bridge with left knee extended (1Lb-LExt) with (**a**) right foot flat (right ankle plantar flexion, PF) and (**b**) right foot up (right ankle dorsiflexion, DF).

**Table 1 bioengineering-11-00356-t001:** Mean EMG ± SD for each muscle and floor bridge exercise for both ankle PF and DF expressed as a percent of each muscle’s maximum isometric voluntary contraction.

Exercise	Ankle Position	VM	VL	RF	TFL	GMED	GMAX	ST	BF	ADD	ES	LATS	RA	EO	IO
2LB	PF	3 ± 1%	2 ± 1%	2 ± 1%	15 ± 8%	14 ± 8%	17 ± 12%	22 ± 8%	18 ± 12%	8 ± 4%	25 ± 7%	14 ± 10%	5 ± 4%	5 ± 5%	7 ± 2%
2LB	DF	4 ± 2%	3 ± 2%	2 ± 1%	15 ± 12%	14 ± 8%	19 ± 14%	14 ± 5%	11 ± 7%	11 ± 4%	25 ± 7%	13 ± 8%	5 ± 4%	6 ± 5%	7 ± 3%
Two-tailed *p*-value		0.191	0.290	0.097	0.792	0.536	0.149	0.0006 *	0.009 *	0.035	0.693	0.477	0.683	0.259	0.786
2LB-ABD	PF	2 ± 1%	2 ± 1%	1 ± 1%	22 ± 16%	21 ± 8%	19 ± 10%	19 ± 8%	16 ± 11%	5 ± 2%	23 ± 8%	13 ± 9%	5 ± 4%	5 ± 4%	6 ± 2%
2LB-ABD	DF	3 ± 4%	2 ± 2%	1 ± 1%	21 ± 12%	22 ± 8%	19 ± 12%	11 ± 7%	11 ± 9%	5 ± 2%	23 ± 8%	14 ± 12%	5 ± 3%	8 ± 6%	6 ± 4%
Two-tailed *p*-value		0.362	0.351	0.606	0.675	0.342	0.619	0.001 *	0.001 *	0.589	0.839	0.494	0.406	0.046	0.827
2LB-ADD	PF	8 ± 5%	5 ± 3%	5 ± 3%	22 ± 16%	20 ± 11%	23 ± 13%	36 ± 12%	30 ± 12%	35 ± 16%	34 ± 10%	21 ± 12%	7 ± 5%	11 ± 9%	14 ± 4%
2LB-ADD	DF	9 ± 5%	6 ± 3%	6 ± 3%	21 ± 14%	18 ± 10%	23 ± 13%	30 ± 10%	26 ± 9%	36 ± 15%	35 ± 14%	23 ± 11%	8 ± 7%	12 ± 9%	14 ± 4%
Two-tailed *p*-value		0.469	0.091	0.116	0.459	0.073	0.977	0.009 *	0.048	0.521	0.318	0.413	0.268	0.233	0.544
1LB-LFlex	PF	8 ± 5%	5 ± 3%	3 ± 2%	64 ± 28%	57 ± 27%	28 ± 11%	38 ± 13%	30 ± 14%	18 ± 6%	33 ± 9%	16 ± 10%	5 ± 3%	8 ± 4%	13 ± 6%
1LB-LFlex	DF	13 ± 7%	10 ± 6%	6 ± 3%	64 ± 28%	58 ± 27%	28 ± 14%	32 ± 14%	25 ± 12%	22 ± 8%	32 ± 7%	17 ± 12%	6 ± 3%	12 ± 4%	17 ± 8%
Two-tailed *p*-value		0.001 *	0.001 *	0.004 *	0.861	0.783	0.821	0.006 *	0.007 *	0.001 *	0.278	0.578	0.200	0.007 *	0.009 *
1LB-LExt	PF	6 ± 4%	5 ± 3%	3 ± 2%	38 ± 17%	41 ± 17%	29 ± 15%	33 ± 12%	29 ± 12%	15 ± 6%	24 ± 7%	14 ± 7%	8 ± 6%	11 ± 8%	10 ± 2%
1LB-LExt	DF	9 ± 5%	9 ± 5%	5 ± 2%	39 ± 18%	41 ± 18%	30 ± 16%	28 ± 11%	24 ± 12%	19 ± 7%	24 ± 9%	15 ± 8%	8 ± 6%	12 ± 8%	11 ± 4%
Two-tailed *p*-value		0.031	0.009 *	0.048	0.553	0.987	0.523	0.009 *	0.004 *	0.001 *	0.785	0.579	0.882	0.145	0.315

* Significant difference (*p* < 0.01) in EMG activity between ankle DF and ankle PL are highlighted and bolded in red. 2LB = two-leg bridge; 2LB-ABD = two-leg bridge with resistance band around knees; 2LB-ADD = two-leg bridge with ball between knees; 1LB-LFlex = one-leg bridge with left knee flexed to chest; 1LB-LExt = one-leg bridge with left knee extended; PF = ankle in plantar flexion with feet flat on floor; DF = ankle in maximum dorsiflexion; VM = vastus medialis; VL = vastus lateralis; RF = rectus femoris; TFL = tensor fascia latae; GMED = gluteus medius; GMAX = gluteus maximus; ST = medial hamstrings (semimembranosus and semitendinosus, but primarily semitendinosus); BF = lateral hamstrings (biceps femoris); ADD = hip adductors (primarily adductor longus); ES = lumbar paraspinals (erector spinae); LATS = latissimus dorsi; RA = upper rectus abdominis; EO = external oblique; IO = internal oblique.

## Data Availability

Data are available from the corresponding author on reasonable request.

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
