# Peer review of "Effects of Ankle Position While Performing One- and Two-Leg Floor Bridging Exercises on Core and Lower Extremity Muscle Recruitment"

_bioengineering, 2024, doi:10.3390/bioengineering11040356_

Round 1
Reviewer 1 Report
Comments and Suggestions for Authors
Thank you for the opportunity to review this study, which examined the effect of two types of foot position (dorsi vs plantar flexion) on various types of bridge movement. I believe more information in the introduction is needed to help form the rationale for the study. It appears to be testing a "commonly held belief" but no rational for this or background information.
The methods need more clarity. order of tests, counter balancing? Some of the bridge movements add resistance (ab or adduction) so is it a repeated measures design?
Stats- paired T-test approach does not seem appropriate for a study set up for an ANOVA, no error corrections (bonferoni) made.
Result- a few tables. No figures. Thgis makes it hard for readers to tease out a pattern of significance- can be improved.
Author Response
Overall Responses to Reviewer 1:
Thank you so much for your excellent recommendations and they have greatly improved our paper. To make it easy for you to find all responses to your specific comments below, in the revised manuscript we included line numbers where changes were made and also we highlighted in yellow the changes we made that corresponded to your recommendations. Thank you!!!
Reviewer 1 Comment:
Thank you for the opportunity to review this study, which examined the effect of two types of foot position (dorsi vs plantar flexion) on various types of bridge movement. I believe more information in the introduction is needed to help form the rationale for the study. It appears to be testing a "commonly held belief" but no rational for this or background information.
Author Response:
You bring up an excellent point. We have expanded our Intro – here is a section of new information we embedded in one of the paragraphs, which now reads as follows in lines 108-126:
Moreover, it is commonly believed that performing these bridging exercises with the feet up and only the heels on the ground (ankle dorsiflexion, DF) recruits the gluteus maximus to a greater extent compared to bridging with the feet flat on the ground (ankle plantarflexion, PF), and bridging with PF recruits the hamstrings to a greater extent compared to bridging with DF [16, 17]. However, these beliefs have never been scientifically validated. The only known study that examined lower extremity muscle activity while bridging with different foot positions was performed by Yoo [18], who examined hamstring and gluteus maximus activity between traditional two leg bridging with the feet flat versus two leg bridging with the heels off the ground. Yoo [18] reported that hamstring activity was significantly less and gluteus maximus activity was significantly greater when bridging while raising the heels off the ground compared to bridging with the feet flat on the ground. However, there are no known studies that have examined hamstring and gluteus maximus activity, as well as additional lower extremity muscles, while bridging on the heels with the toes up (DF ankle position) instead of on the toes with the heels up. Therefore, the purpose of this study was to assess how employing two different ankle positions (PF versus DF) while performing five common bridging exercises (3 bipedal and 2 unipedal) [16, 17] used in rehabilitation and athletic performance affect core and select lower extremity muscle EMG recruitment.
Reviewer 1 Comment:
The methods need more clarity. order of tests, counter balancing? Some of the bridge movements add resistance (ab or adduction) so is it a repeated measures design?
Author Response:
Sorry we were not more clear here and “yes” to your questions. To enhance clarity in our Methods as you asked, we added the following to our methods in lines 281-285:
All MVIC’s and exercises were first randomized, and then to counterbalance the repeated measures design and minimize the risk of an order effect, half of the subjects performed all MVIC’s and exercises in the randomized order and the other half of the subjects performed all MVIC’s and exercises in the reverse order.
Reviewer 1 Comment:
Stats- paired T-test approach does not seem appropriate for a study set up for an ANOVA, no error corrections (Bonferroni) made.
Author Response:
We originally chose the ANOVA and examined both the effect of ankle position (dorsiflexion versus plantar flexion) and exercise type, but when we originally submitted the reviewers said it was too much data and convoluted to present that much and recommended we simplify and simply compare the effect of one ankle position to the other ankle position (paired t-test since all 20 subjects performed both ankle positions). You bring up a great point regarding no error correction for multiple tests so what we would like to do with your approval is change our p-value from p<0.05 to the more stringent p<0.01. We changed our methods, results, and Discussion accordingly. Thank you for bringing this to our attention.
Reviewer 1 Comment:
Result- a few tables. No figures. This makes it hard for readers to tease out a pattern of significance- can be improved.
Author Response:
Thank you for this. As you requested and in order to make Table 1 and significant differences easier to locate, we highlighted and bold in red all significant differences in Table 1. Just below Table 1 in the caption we revised as follows and throughout Table 1 we added “%” so the units are now included:
Table 1. Mean EMG ± SD for each muscle and floor bridge exercise for both ankle PF and DF expressed as a percent of each muscle’s maximum isometric voluntary contraction.
*Significant difference (p < 0.01) in EMG activity between ankle DF and ankle PL are highlighted and bolded in red.

Reviewer 2 Report
Comments and Suggestions for Authors
Dear authors,
I would like to express my gratitude regarding the opportunity to review this manuscript.
At this stage the manuscript requires several improvements. Below suggestions with line indication:
Line 86. Justify why you chose such exercise for research? Why did you choose this group for the research? Why were they women and men? Wouldn't it be better to use a homogeneous group? Have you considered the effect of fat tissue on EMG signal attenuation?
Line 196. How was the order of exercises randomized in these studies?
Line 205. Why was the T test chosen? Has the normality of distributions been tested?
Line 208. Explain the observed statistical differences between normalized muscle activities since they are less than the standard deviation. Don't you think there is something wrong with this analysis? For example, BF 2LB PF = 18±12 and BF 2LB DF = 11±7 and p=0.009, the difference is only 7 and the standard deviations are from 7 to 12. These differences are smaller than the standard deviations. Additionally, for better clarity, these results can be presented as a percentage of the difference from the MVIC value. If you present them this way, why are there no units? It also follows from these data that the activity of selected muscles is very low relative to MVIC. This means you chose the wrong exercises or muscles. The variability of results in a group may be random and does not depend on the position of the ancle. How can you justify the point of analyzing such small values when they are close to the noise of EMG signal?
In my opinion, if anything, the results could have been compared based on the exercises but not the position of the ankle. This could have been added as an additional variable. It is worth considering how to analyze such small values
Discussion and conclusions. Comments after receiving answers to the above questions.
Author Response
Overall Responses to Reviewer 2:
Thank you so much for your excellent recommendations and they have greatly improved our paper. To make it easy for you to find all responses to your specific comments below, in the revised manuscript we included line numbers where changes were made and also we highlighted in yellow the changes we made that corresponded to your recommendations. Thank you!!!
Reviewer 2 Comment:
Line 86. Justify why you chose such exercise for research? Why did you choose this group for the research? Why were they women and men? Wouldn't it be better to use a homogeneous group? Have you considered the effect of fat tissue on EMG signal attenuation?
Author Response:
All excellent questions! We chose the exercises based on what exercises are commonly done both rehabilitation of the trunk and lower extremity and based on what the fact these are common exercises used by athletes. We have both physical therapists and trainers/coaches of athletes as authors on the paper which allows us to know what exercises are commonly used both in the clinic and by athletes. Given all these bridging exercises are commonly used by males and females, and the fact that we normalized the EMG by MVIC’s, that was our justification of including both males and females. While the normalization process does drastically help with attenuation on the EMG signal (ie, low signal during exercises will also produce low signal during MVIC, which offset each other so in theory normalized EMG should no be drastically affect by higher and lower subcutaneous fat), nevertheless we also only use subjects normal or below normal amounts of body fat and we always assess body fat to confirm this. While we included a statement regarding this is other similar studies we published in the literature, we will not include it in this paper and we should have. Therefore, we added the following statement in our Methods in lines 135-141:
To optimize the quality of the electromyographic (EMG) signal collected, this study was limited to a convenience sample of 20 healthy, young subjects (10 male and 10 female) who had normal or below normal body fat for their age group, in accordance with standards set by the American College of Sports Medicine [19]. Baseline skinfold calipers (Model 68900, Country Technology, Inc., Gays Mill, WI) and appropriate regression equations were used to assess percent body fat. Mean(SD) percent body fat were 18.2(2.4)% for females and 11.7(3.1)% for males.
Reviewer 2 Comment:
Line 196. How was the order of exercises randomized in these studies?
Author Response:
Thank you for this as we should have been more clear on how we randomized the order of testing so we could prevent an order effect. We added a statement in the Methods to clear this up, and it reads as follows in lines 281-285:
All MVIC’s and exercises were first randomized, and then to counterbalance the repeated measures design and minimize the risk of an order effect, half of the subjects performed all MVIC’s and exercises in the randomized order and the other half of the subjects performed all MVIC’s and exercises in the reverse order.
Reviewer 2 Comment:
Line 205. Why was the T test chosen? Has the normality of distributions been tested?
Author Response:
We originally chose the ANOVA and examined both the effect of ankle position (dorsiflexion versus plantar flexion) and exercise type, but when we originally submitted the reviewers said it was too much data and convoluted to present that much and recommended we simplify and simply compare the effect of one ankle position to the other ankle position (paired t-test since all 20 subjects performed both ankle positions). Yes, we did test for a normal distribution.
Reviewer 2 Comment:
Line 208. Explain the observed statistical differences between normalized muscle activities since they are less than the standard deviation. Don't you think there is something wrong with this analysis? For example, BF 2LB PF = 18±12 and BF 2LB DF = 11±7 and p=0.009, the difference is only 7 and the standard deviations are from 7 to 12. These differences are smaller than the standard deviations. Additionally, for better clarity, these results can be presented as a percentage of the difference from the MVIC value. If you present them this way, why are there no units? It also follows from these data that the activity of selected muscles is very low relative to MVIC. This means you chose the wrong exercises or muscles. The variability of results in a group may be random and does not depend on the position of the ancle. How can you justify the point of analyzing such small values when they are close to the noise of EMG signal?
Author Response:
We double checked these and they are correct using the paired t-test. Part of the explanation we believe is the fact it is a repeated measures design and also the relatively high number of subjects makes a significant difference even though the change between two values may not be very big. The low values show that some of the muscles are not very active at all but there are numerous values that show moderate to higher activity relative to the MVIC, such as 64±28%, 57±27%, 41±17%, 38±17%, 41±18%, 39±18%, 36±12%, 35±15%, 34±16%, 34±10%, 35±14%, 30±16%, 30±10%, 29±15%, 28±11%, 28±14%, etc… Also, as you requested we highlighted and bold in red all significant differences in Table 1 to make it easier to locate, we added % to each value in Table 1 instead of just saying they were expressed as a percent, and just below Table 1 in the caption we revised as follows to make more clear:
Table 1. Mean EMG ± SD for each muscle and floor bridge exercise for both ankle PF and DF expressed as a percent of each muscle’s maximum isometric voluntary contraction.
*Significant difference (p < 0.01) in EMG activity between ankle DF and ankle PL are highlighted and bolded in red.
Reviewer 2 Comment:
In my opinion, if anything, the results could have been compared based on the exercises but not the position of the ankle. This could have been added as an additional variable. It is worth considering how to analyze such small values
Author Response:
That is how we first wrote the paper and submitted it (to a different journal) and used the 2-way repeated measures ANOVA (ankle position and exercise type as the two independent variables) and all three reviewers said there was too much information that convoluted the results and if ankle position was the most important variable to us (and it is), then focus just on the difference between the two ankle positions for each exercise. Therefore, at this point we would like to request we keep it as is with ankle position being the only variable we are examining among the exercises.
Reviewer 2 Comment:
Discussion and conclusions. Comments after receiving answers to the above questions.
Author Response:
OK, thank you!!!

Reviewer 3 Report
Comments and Suggestions for Authors
The submitted manuscript, titled "Effects of ankle position while performing one and two leg floor bridging exercises on core and lower extremity muscle recruitment," is overall well-structured and provides a comprehensive and up-to-date list of citations on the topic. However, there are some critical points that the Authors should address to enhance the quality of the work. The following, are my concerns:
-
Abstract: In the context of the Abstract, it is suggested to emphasize the context and innovation of the research compared to existing literature. The detailed list of analyzed muscles could be reduced, focusing on the most innovative aspects of the study;
-
Line 83: Specify in more detail the parameters that will be used to evaluate the electrical activity of muscles during the two exercise conditions;
-
Participant Selection: I was wondering if the presence of previous orthopedic pathologies was assessed during the participant selection for the study. Please clearly state the exclusion/inclusion criteria in the manuscript;
-
Feedback on Execution Quality: Feedback on the quality of exercise execution was provided only during the pre-test or also during the acquisitions? Why not perform the pre-test immediately before the acquisition sessions?
-
Captions in images: Include titles and captions in the images to improve readability (e.g., including muscle labels, exercise names, and execution conditions);
-
Temporal segmentation of exercises: Explain how the exercises were temporally segmented, whether manually through video or through another method;
-
Time Between Pre-test and Acquisitions: Explain the reason for the one week between the pre-test and acquisitions, considering that volunteers may have forgotten the correct instructions;
-
EMG probe attachment: Verify the discrepancy between what is stated in the text and the images regarding the attachment of EMG probes' wires to the volunteers' skin.
-
EMG pre-processing analysis: Clearly indicate the programming language and environment used to perform EMG signal pre-processing analyses.
-
Statistical tests: Explain the rationale behind choosing the t-test for comparing normalized EMG data and indicate whether the normality of data distribution was tested. Many other statistical tests are available for time-series assessment (e.g., Statistical Parametric Mapping, etc.);
-
Average EMG signals: I warmly suggest the inclusion of a representation of average EMG signals;
-
Table clarity: Please improve the clarity of the table, reorganizing the data, highlighting significant comparisons in bold, and including missing units of measurement;
-
Data variability: Please comment on data variability, especially in cases where variability is significant.
-
Standards for EMG probe placement: Please provide details on the standards followed for the placement of EMG probes and explain how the quality of signals and the level of crosstalk were assessed.
Best regards
Author Response
Reviewer 3 General Comment:
The submitted manuscript, titled "Effects of ankle position while performing one and two leg floor bridging exercises on core and lower extremity muscle recruitment," is overall well-structured and provides a comprehensive and up-to-date list of citations on the topic. However, there are some critical points that the Authors should address to enhance the quality of the work. The following, are my concerns:
Overall Responses to Reviewer 3:
Thank you so much for your excellent recommendations and they have greatly improved our paper. To make it easy for you to find all responses to your specific comments below, in the revised manuscript we included line numbers where changes were made and also we highlighted in yellow the changes we made that corresponded to your recommendations. Thank you!!!
Reviewer 3 Comment:
Abstract: In the context of the Abstract, it is suggested to emphasize the context and innovation of the research compared to existing literature. The detailed list of analyzed muscles could be reduced, focusing on the most innovative aspects of the study;
Author Response:
We have done/attempted to do as you requested although there is not any literature related to ankle position with bridging for multiple lower extremity musculature. We shortened the description of the muscles analyzed and also shortened the findings. See lines 15-27.
Reviewer 3 Comment:
Line 83: Specify in more detail the parameters that will be used to evaluate the electrical activity of muscles during the two exercise conditions;
Author Response:
We looked at line 83 as you specified and it says “Strengthening the lumbopelvic-hip complex has been shown to decrease lower extremity”. Did you perhaps mean in the abstract??? We will go with that assumption and we added in the abstract the “Noraxon Telemyo Direct Transmission System” in line 23.
Reviewer 3 Comment:
Participant Selection: I was wondering if the presence of previous orthopedic pathologies was assessed during the participant selection for the study. Please clearly state the exclusion/inclusion criteria in the manuscript;
Author Response:
Related to inclusion/exclusion criteria we revised the manuscript to now read as follows in lines 143-148:
Individuals were excluded from the study if they had any musculoskeletal pathologies, as assessed by a licensed physical therapist, that prevented them from being able to perform all exercises pain-free, through their full range of motion, and with proper form and technique. All subjects were also excluded from the study if they did not have at least 3 years’ experience in performing bipedal and unipedal floor bridging exercises, and were excluded if their body fat was above normal, as described previously.
Reviewer 3 Comment:
Feedback on Execution Quality: Feedback on the quality of exercise execution was provided only during the pre-test or also during the acquisitions? Why not perform the pre-test immediately before the acquisition sessions?
Author Response:
Feedback, if needed, was provided during both pre-test and during data acquisitions; however, since all subjects were experienced in all bridging exercises feedback was seldom needed as exercises were performed correctly. Just to make sure they did one practice rep before every data acquisition. Regarding why the pre-test and data acquisitions were on different days:
There are several reasons why we made the pre-test 1 week before the actual data collection, and we have clarified this better in the paper. In the pre-test each participant was examined by a licensed Physical Therapist to make sure they had no musculoskeletal pathology that prevented them from executing each exercise with proper form and technique. We also made sure they could perform each exercise, so it was a practice session (but these participants all had at least 3 years’ experience in performing unipedal and bipedal bridging exercises. Lastly we also assessed their body fat and we have now described this in the manuscript. Therefore, the pre-test itself took a significant amount of time and effort and we didn’t want fatigue to be a factor, or a data collection session to be excessively long.
Reviewer 3 Comment:
Captions in images: Include titles and captions in the images to improve readability (e.g., including muscle labels, exercise names, and execution conditions);
Author Response:
We wonder if somehow the captions did not appear in the version of our manuscript you received, because in our version we included a caption with the exercise names and also shows both conditions for each exercise. Every image (figure) has its own caption. For example, the Figure captions for all 5 exercises says the following:
FIGURE 1a (top) and1b (bottom). Two-leg bridge (2LB) with a) both feet flat (ankle plantar flexion, PF) and b) both feet up (ankle dorsiflexion, DF).
FIGURE 2a (top) and 2b (bottom). Two-leg bridge with resistance band around knees (2LB-ABD) with a) both feet flat (ankle plantar flexion, PF) and b) both feet up (ankle dorsiflexion, DF)
FIGURE 3a (top) and 3b (bottom). Two-leg bridge with ball between knees (2LB-ADD) with a) both feet flat (ankle plantar flexion, PF) and b) both feet up (ankle dorsiflexion, DF)
FIGURE 4a (top) and 4b (bottom). One-leg bridge with left knee flexed to chest (1LB-LFlex) with a) right foot flat (right ankle plantar flexion, PF) and b) right foot up (right ankle dorsiflexion, DF)
FIGURE 5a (top) and 5b (bottom). One-leg bridge with left knee extended (1LB-LExt) with a) right foot flat (right ankle plantar flexion, PF) and b) right foot up (right ankle dorsiflexion, DF)
The only thing not in the caption that you are requesting is muscle labels. We are not positive what you are asking for here, but if it is the muscles, we added the following at the end of each of the figures above:
EMG surface electrodes were positioned over the following muscles: vastus medialis; vastus lateralis; rectus femoris; tensor fascia latae; gluteus medius; gluteus maximus; medial hamstrings (semimembranosus and semitendinosus, but primarily semitendinosus); lateral hamstrings (biceps femoris); hip adductors (primarily adductor longus); lumbar paraspinals (erector spinae); latissimus dorsi; upper rectus abdominis; external oblique; internal oblique
Reviewer 3 Comment:
Temporal segmentation of exercises: Explain how the exercises were temporally segmented, whether manually through video or through another method;
Author Response:
EMG data were collected during a 5sec isometric contraction during the end positions shown for each exercise in Figures 1-5. Once the subject was in the “ending” position, the subject stationary stayed in that position until a tester told the subject to relax. The Noraxon Telemyo Direct Transmission System was pre-set for a 5 sec data collection period. As soon as the subject assumed the ending stationary position Data Collection started and it automatically ended 5 sec later. After it ended a tester instructed the subject to relax.
Reviewer 3 Comment:
Time Between Pre-test and Acquisitions: Explain the reason for the one week between the pre-test and acquisitions, considering that volunteers may have forgotten the correct instructions;
Author Response:
There are several reasons why we made the pre-test 1 week before the actual data collection, and we have clarified this better in the paper. In the pre-test each participant was examined by a licensed Physical Therapist to make sure they had no musculoskeletal pathology that prevented them from executing each exercise with proper form and technique. We also made sure they could perform each exercise, so it was a practice session (but these participants all had at least 3 years’ experience in performing unipedal and bipedal bridging exercises. Lastly we also assessed their body fat and we have now described this in the manuscript. Therefore, the pre-test itself took a significant amount of time and effort and we didn’t want fatigue to be a factor, or a data collection session to be excessively long.
Reviewer 3 Comment:
EMG probe attachment: Verify the discrepancy between what is stated in the text and the images regarding the attachment of EMG probes' wires to the volunteers' skin.
Author Response:
This was our mistake. In the original we referenced an older Noraxon Myosystem we had used previously in some of our published work but for this study we used a newer Noraxon Telemyo Direct Transmission System which directly transmits data from the EMG electrode site to a belt-worn receiver, which can be seen in the figures. We corrected this so Lines 23 and 270 now say “Noraxon Telemyo Direct Transmission System”. Thank you!!!!!
Reviewer 3 Comment:
EMG pre-processing analysis: Clearly indicate the programming language and environment used to perform EMG signal pre-processing analyses.
Author Response:
Thank you for this. We added the following to the manuscript in 287-293:
Raw EMG signals were processed using Noraxon Myoresearch (Noraxon USA, Inc., Scottsdale, AZ) and were full-waved rectified, smoothed with a 10 ms moving average window and linear enveloped, and then averaged over the entire 5sec duration of each exercise performed. For each repetition, the EMG data were normalized for each muscle and expressed as a percentage of a subject's highest corresponding MVIC trial, which was determined by calculating throughout the 5sec MVIC the highest average EMG signal over a 1sec time interval. Normalized EMG data were then used in statistical analyses.
Reviewer 3 Comment:
Statistical tests: Explain the rationale behind choosing the t-test for comparing normalized EMG data and indicate whether the normality of data distribution was tested. Many other statistical tests are available for time-series assessment (e.g., Statistical Parametric Mapping, etc.);
Author Response:
We originally chose the ANOVA and examined both the effect of ankle position (dorsiflexion versus plantar flexion) and exercise type, but when we originally submitted the reviewers said it was too much data and convoluted to present that much and recommended we simplify and simply compare the effect of one ankle position to the other ankle position (paired t-test since all 20 subjects performed both ankle positions). Yes, we did test for a normal distribution.
Reviewer 3 Comment:
Average EMG signals: I warmly suggest the inclusion of a representation of average EMG signals.
Author Response:
OK, we have a bit of a problem with this one. It has been a few years since we collected our data and our Noraxon system since our study has “died”, like old systems do eventually, and we replaced it with a Delsys EMG system. BUT I will tell you that because each exercise was performed “isometrically” with no movement and with a constant intensity and effort, the average EMG signal was consistent and stable from time 0 sec to time 5 sec when we originally collected the data. So sorry!!!
Reviewer 3 Comment:
Table clarity: Please improve the clarity of the table, reorganizing the data, highlighting significant comparisons in bold, and including missing units of measurement;
Author Response:
Thank you for this. As you requested we highlighted and bold in red all significant differences in Table 1 to make it easier to locate, we added % to each value in Table 1 instead of just saying they were expressed as a percent, and just below Table 1 in the caption we revised as follows to make more clear:
Table 1. Mean EMG ± SD for each muscle and floor bridge exercise for both ankle PF and DF expressed as a percent of each muscle’s maximum isometric voluntary contraction.
*Significant difference (p < 0.01) in EMG activity between ankle DF and ankle PL are highlighted and bolded in red.
Reviewer 3 Comment:
Data variability: Please comment on data variability, especially in cases where variability is significant.
Author Response:
The variability of our EMG data is similar to the variability of EMG data from all the other bridging studies in the literature.
Reviewer 3 Comment:
Standards for EMG probe placement: Please provide details on the standards followed for the placement of EMG probes and explain how the quality of signals and the level of crosstalk were assessed.
Author Response:
As we reported in our manuscript, potential limitations of this study include the potential for cross talk from neighboring muscles from surface EMG. However, careful electrode placement that adhered to established protocols was employed to mitigate such outcomes, and the electrode placements utilized in our manuscript have been shown to minimize EMG cross-talk from other muscles. The references for these statements are as follows:
McGill S, Juker D, Kropf P. Appropriately placed surface EMG electrodes reflect deep muscle activity (psoas, quadratus lumborum, abdominal wall) in the lumbar spine. J Biomech. 1996; 29(11): 1503-7.
Ng JK, Kippers V, Richardson CA. Muscle fibre orientation of abdominal muscles and suggested surface EMG electrode positions. Electromyogr Clin Neurophysiol. 1998; 38(1): 51-8.
Consistent with these references, we also used electrode placement techniques to minimize cross talk as recommended by Basmajian J, Blumenstein R, Electrode Placement in EMG Biofeedback. 1980, Baltimore, MD: Williams and Wilkins. 79-86. Basmajian is considered by many as the “father of EMG Biofeedback”

Round 2
Reviewer 1 Report
Comments and Suggestions for Authors
The revisions are adequate for publication. I still am not satisfied by only a large table and feel that key muscles of interest could at least be shown in a figure. One does not need to put everything in a single figure just like a single table is also not the best.
Reviewer 2 Report
Comments and Suggestions for Authors
Suggested corrections have been introduced and the article is ready for publication